# Anti-Herpetic Activity of Killer Peptide (KP): An In Vitro Study

**DOI:** 10.3390/ijms251910602

**Published:** 2024-10-01

**Authors:** Arianna Sala, Francesco Ricchi, Laura Giovati, Stefania Conti, Tecla Ciociola, Claudio Cermelli

**Affiliations:** 1Department of Surgical, Medical, Dental and Morphological Sciences with Interest in Transplant, Oncological and Regenerative Medicine, University of Modena and Reggio Emilia, 41121 Modena, Italy; sala.arianna@aou.mo.it (A.S.); francesco.ricchi@unimore.it (F.R.); 2Clinical and Experimental Medicine Ph.D. Program, University of Modena and Reggio Emilia, 41121 Modena, Italy; 3Laboratory of Microbiology and Virology, Department of Medicine and Surgery, University of Parma, 43125 Parma, Italy; laura.giovati@unipr.it (L.G.); stefania.conti@unipr.it (S.C.); tecla.ciociola@unipr.it (T.C.); 4Microbiome Research Hub, University of Parma, 43124 Parma, Italy

**Keywords:** antimicrobial peptides, HSV-1, HSV-2, antivirals, Killer Peptide (KP)

## Abstract

Antimicrobial peptides represent a promising alternative to traditional drugs in relation to cost, toxicity, and, primarily, the growing problem of drug resistance. Here, we report on the activity against HSV-1 and HSV-2 of a previously described wide-spectrum synthetic decapeptide, Killer Peptide (KP). As determined by plaque reduction assays, treatment with KP at 100 μg/mL resulted in a reduction in the viral yield titer of 3.5 Logs for HSV-1 and 4.1 Logs for HSV-2. Further evaluation of KP antiviral activity focused on the early stages of the virus replicative cycle, including the determination of the residual infectivity of viral suspensions treated with KP. A direct effect of the peptide on viral particles impairing virus absorption and penetration was shown. The toxicity profile proved to be extremely good, with a selectivity index of 29.6 for HSV-1 and 156 for HSV-2. KP was also active against acyclovir (ACV)-resistant HSV isolates, while HSV subcultures in the presence of sub-inhibitory doses of KP did not lead to the emergence of resistant strains. Finally, the antiviral action of KP proved to be synergistic with that of ACV. Overall, these results demonstrate that KP could represent an interesting addition/alternative to acyclovir for antiviral treatment.

## 1. Introduction

Herpes Simplex Viruses type 1 (HSV-1), the main cause of oral herpes, and type 2 (HSV-2), the main cause of genital herpes, are globally widespread and represent a significant health problem. HSV-1 infection usually occurs early in life through the oral mucosa, while HSV-2 is predominantly acquired through sexual contact. However, a growing body of evidence from Europe, North America, and Asia also suggests increasing sexual transmission for HSV-1 [1,2,3,4,5]. HSV-1 is the most widespread, with seroprevalence values ranging from 65% to >90% and a higher circulation in low-income countries. An estimated 3.7 billion people under age 50 (67%) globally have HSV-1 infection, whereas 491 million people aged 15–49 (13%) worldwide have HSV-2 infection. Both viruses usually cause asymptomatic or paucisymptomatic infections but can be associated with severe or life-threatening diseases, especially in immunocompromised hosts or following vertical transmission. Encephalitis, hepatitis, and pneumonia are common events in transplanted patients with HSV reactivation [6,7]. Noteworthy, HSV-1 reactivation with a high pathogenic impact has been reported in a significant percentage of patients during the ongoing COVID-19 pandemic [8]. Both viruses cause neonatal herpes infection, with systemic dissemination, always severe and even fatal. Reported incidence rates range from 1.6/100,000 to 8.4/100,000 live births.

Currently, acyclovir (ACV) is the first-choice drug for the treatment of both HSV infections, but other effective drugs are available, including vidarabine, trifluridine, ganciclovir, foscarnet, and cidofovir. Except for foscarnet, all these agents are nucleoside analogs that are incorporated into the growing viral DNA chain and interfere with viral replication. Foscarnet is a pyrophosphate that directly inhibits viral DNA polymerase. However, prolonged drug use for the treatment of recurrent herpetic infections in immunocompromised patients has led to the emergence of resistant viral strains. Moreover, the treatment of congenital infections in newborns is extremely difficult due to drug toxicity. Therefore, new therapeutic approaches are urgently demanded, among which antimicrobial peptides are highly promising.

We previously investigated the inhibitory activity of a set of 13 peptides against four viruses (namely, HSV-1, Adenovirus, Vesicular Stomatitis Virus, and Coxsackievirus B5 [9]. In particular, the decapeptide named KP (Killer Peptide) showed the highest antiviral activity. KP derives from the sequence of the variable region of a single-chain recombinant anti-idiotypic antibody representing the internal image of a yeast killer toxin characterized by a wide spectrum of antimicrobial activity [10]. KP has previously been proven to inhibit HIV and Influenza A virus replication through different mechanisms [11]. The aim of the present work was to characterize the activity of KP against HSV-1 and HSV-2 in terms of (1) viral yield inhibition, (2) timing of action during the viral replication cycle, (3) activity on ACV-resistant isolates, (4) synergistic vs. additive effect with ACV, and (5) capability to select KP-resistant strains.

## 2. Results

### 2.1. Cytotoxicity Assay on Vero Cells

The MTT assay was used to determine the cytotoxicity on Vero cells of the three investigated molecules, i.e., KP, SP (a decapeptide containing the same residues of KP with a different sequence), and ACV. None of them caused a significant reduction in cell viability, in comparison with the untreated controls, at the highest concentration of use (100 μg/mL). The reduction in cell viability was always <15% (Figure 1). Half maximal cytotoxic concentration values (CC_50_) were then calculated (see Table 1 below).

### 2.2. Peptide Antiviral Activity

In order to evaluate the antiviral potential of KP against HSV-1 and HSV-2, viral yield reduction assays were performed. SP was used as a control, and each peptide was tested at three concentrations (10, 50, and 100 μg/mL). The results, expressed as logarithm of the viral yield reduction in treated samples compared with the negative (untreated) controls, are shown in Figure 2A. Regarding the activity against HSV-1, KP showed considerable inhibition at 50 and 100 μg/mL, with a reduction of 2.7 and 3.5 Logs, respectively. On the other hand, KP showed lower antiviral activity against HSV-2 at the concentration of 50 μg/mL (1.2 Log reduction) but a higher inhibitory activity at the maximum concentration (4.1 Log reduction). At the concentration of 10 μg/mL, there was no statistically significant reduction in viral load for both viruses in comparison to SP treatment. SP, at all the assayed concentrations, did not show a reduction greater than 0.2 Log compared with the negative (untreated) controls.

Furthermore, the viral yield was quantified by real-time PCR to confirm the results obtained with the biological assays. Figure 2B shows the data obtained from the relative quantification (RQ) of the viral genome in the samples treated with KP at 100 μg/mL compared to the negative (untreated) controls. A very low RQ was obtained for HSV-1 (0.035) and HSV-2 (0.0001) after KP treatment.

### 2.3. Time-of-Drug Addition Assay

In order to establish the time point during the virus replication cycle of maximal antiviral activity, a time-of-drug addition assay was performed. The graph in Figure 3 shows that KP antiviral activity has the same trend for both viruses. The major inhibition was circumscribed to the early stages of the replicative cycle. KP addition concomitantly with the viral infection (time 0) induced a viral yield reduction of 3.4 Logs in HSV-1 and 4 Logs in HSV-2. When the peptide was added 1, 3, 6, 12, and 18 h post-infection, the viral load reduction did not exceed 0.7 Log for both viruses.

### 2.4. Characterization of KP Antiviral Activity

For further evaluation of KP antiviral activity, the investigation focused on the early stages of the virus replicative cycle, including pretreatment of either cell-free HSV suspension or Vero cell monolayer prior to infection. Figure 4 shows the overall results obtained with different approaches on HSV-1 and HSV-2. Vero cells monolayer pretreatment with KP 1 h before infection showed no efficacy against both viruses. Viral yield after direct exposition of HSV-1 suspension to the peptide for 1 h before infection showed a statistically significant reduction of 2.3 Logs, i.e., a virucidal activity, whereas HSV-2 viral yield reduction was about 1 Log. In the assays of inhibition of adsorption and penetration, the viral yield was reduced by 1.3 and 0.3 Logs, respectively, for HSV-1, while for HSV-2, the reduction was 1.7 and 0.9 Logs, respectively.

### 2.5. KP and ACV Inhibitory Concentration 50 (IC_50_) and Selectivity Index (SI)

Considering the antiviral activity displayed by KP against HSVs, the IC_50_ and SI (SI = CC_50_/IC_50_) were determined. Table 1 reports the results of the comparison with ACV. CC_50_ values were 401.6 μg/mL for KP and 862.5 μg/mL for ACV, confirming a very low toxicity for the investigated compounds. KP IC_50_ value was 13.6 μg/mL against HSV-1 and 2.6 μg/mL against HSV-2, while ACV IC_50_ values were 0.5 μg/mL and 1.3 μg/mL, respectively. Therefore, KP showed a lower efficacy in comparison with ACV on both viruses. Anyway, the antiviral activities of KP and ACV were highly remarkable, as disclosed by SI values (29.6 and 1742.5 for HSV-1 with KP and ACV, respectively; 156 and 669.1 for HSV-2 with KP and ACV, respectively).

### 2.6. KP Antiviral Activity against ACV-Resistant HSV Isolates

KP anti-herpetic activity was also evaluated against HSV-1 and HSV-2 ACV-resistant isolates by a viral yield reduction assay (Figure 5). The viral yield reduction in wild-type strains following ACV and KP treatment was 3.4 and 3.1 Logs for HSV-1 (panel A, left) and 3.9 and 4 Logs for HSV-2 (panel B, left), respectively.

Following ACV treatment, the HSV-1 and HSV-2 ACV-resistant isolates showed 0.6 and 0.3 Log reductions, respectively, whereas after KP treatment, the viral yields of the ACV-resistant isolates were comparable to those of the wild-type strains (3.2 and 3.7 Log reduction in HSV-1 and HSV-2, respectively) (Figure 5, panels A and B, right).

### 2.7. Selection by KP of Resistant Mutants

Following exposure to sub-inhibitory concentrations of KP (10 μg/mL) up to 20 subcultures, both viruses did not show any reduction in sensitivity to inhibitory KP concentrations. As a matter of fact, viral yield experiments carried out in passages 10 and 20 (after culturing in the presence of KP) displayed the same reduction values as the initial isolates. For HSV-1 viral production, these values ranged between 3.5 Log and 3.7 Log in any case, while for HSV-2, they ranged between 3.9 Log and 4.1 Log, with no statistical differences.

### 2.8. Synergy Studies

The potential synergy of KP with ACV was assessed using different concentrations with the same KP/ACV ratio (1:1). Using the CompuSyn software, the combination index (CI) was determined in order to ascertain synergism (CI < 1), antagonism (CI > 1), or additive effect (CI = 1). The CI values are displayed in Table 2, and the compound interaction was characterized by fractional inhibition (Fi)-CI Plots, as shown in Figure 6. All the tested combinations revealed a synergistic interaction, though with different extents.

The advantage of a synergistic drug combination is the reduction in the dose of each drug, which reduces the toxicity and cost of the therapy while maintaining the same efficacy. This concept is expressed as dose-reduction indexes (DRIs), which were 15 and 2570 for ACV and KP, respectively, against HSV-1 and 102 and 376, respectively, for HSV-2 (Table 2).

## 3. Discussion

Antimicrobial peptides (AMPs) represent an interesting new class of anti-infectious agents being developed as alternative or additional therapeutic options to conventional drugs [12,13]. These small molecules have the ability to act on multiple targets and are, therefore, characterized by a wide spectrum of antimicrobial activity against bacteria, yeasts, fungi, and viruses.

KP is a decapeptide with a well-documented wide range of antimicrobial activity [10], including antiviral action against HIV and Influenza A virus by different mechanisms of action [11]. We previously studied the antiviral activity of 13 AMPs, including KP, against different RNA (Coxsackievirus B5, Vesicular Stomatitis Virus) and DNA (HSV-1, Adenovirus-5) viruses. KP showed the highest viral inhibition [9]. Therefore, the purpose of this study was to characterize KP activity against HSV-1 and HSV-2 and its possible mechanisms of action more deeply. Moreover, a comparison with the first-choice drug for HSV infection, i.e., ACV, was carried out.

HSV infections are very common and cause human diseases from mild to fatal, especially in immunocompromised or elderly individuals. Long-term treatments of immunocompromised patients with ACV and its derivatives are leading to the selection of drug-resistant isolates that prompt the research of alternative therapeutic approaches.

Firstly, we investigated the antiviral potential of KP against HSV-1 and HSV-2 by viral yield reduction assays. A remarkable dose-dependent inhibition of the replication of both viruses was observed (3.7 Log for HSV-1, 4.1 Log for HSV-2 at 100 µg/mL) with high SI values (29.6 for HSV-1 and 156 for HSV-2) (Table 1). These results were confirmed by qPCR. Although KP showed a lower efficacy on both viruses, in comparison to ACV, interestingly, the two molecules showed a synergistic effect in any combination tested (Figure 6, Table 2). This result is appealing, so a therapy employing both of them could allow us to reduce the dose of each, with great benefits in terms of toxicity and saving money. Moreover, a very remarkable finding is the unmodified inhibitory activity of KP against ACV-resistant viral isolates (Figure 5), which is a considerable advantage because of the increase in the circulation of ACV-resistant mutants. Moreover, differently from ACV, KP does not select resistant strains. In fact, after several sub-cultures in the presence of sub-inhibitory concentrations of KP, neither virus showed a reduction in sensitivity. Overall, though less active than ACV, KP seems to represent a good therapeutic alternative to the first-choice drug, alone or in combination.

Second, we carried out investigations to disclose the KP mechanism(s) of action. Time-course experiments were performed to identify the step(s) of the virus replication cycle affected by KP. Both HSVs were inhibited only during the very early stages of their replication, within 1 h post-infection (Figure 3). Therefore, we investigated the effects of KP on virus adsorption and penetration. The two assays were carried out at different temperatures, 4 °C and 37 °C, allowing us to discriminate which of the two events was affected by KP: indeed, virus adsorption is a passive step not requiring cell metabolic activity, so it can also occur at 4 °C, while virus entry needs an active involvement of the cell machinery, and 37 °C temperature is necessary. For both viruses, adsorption was a more affected step (1.3 Log reduction in HSV-1, 1.73 in HSV-2). The viral yield was only slightly reduced (0.3 Log, 0.9 Log for HSV-1 and HSV-2, respectively) when KP was added after virus adsorption. We also investigated whether KP could display a direct virucidal activity, i.e., irreversibly damage virions before their attachment to the target cells. In this case, the virus replication was reduced by 2.3 Logs and 1 Log for HSV-1 and HSV-2, respectively. The combination of this virucidal activity, which could likely occur outside the cells before virus interaction with its cell receptor, and inhibition of virus attachment and penetration (the latter with a minor role, especially for HSV-1) could account for the antiviral properties of KP.

Many AMPs perform their activity through membrane destabilization [14]. Membrane stability is highly influenced by its composition in lipids, which has a positive correlation with cholesterol content. Consequently, human cell membranes, particularly rich in cholesterol are less sensitive to the destabilizing activity of antimicrobial peptides in comparison with fungi and bacteria. Similarly to human cell membranes from which HSVs bud, cholesterol is abundant in their envelope, suggesting that the antiviral activity depends more on interference with virus–cell interactions than envelope disruption. Virus attachment to the target cells depends on the glycoprotein composition, with an important role played by ionic charges, while lipids are involved in virus entry by membrane fusion; interestingly, antimicrobial peptides can interfere with membrane charges [15,16,17,18]. HSV-1 and HSV-2, although very similar, have a different set of envelope glycoproteins accounting for the differences observed in their interactions with the host cell. An example is given by the two most representative glycoproteins that mediate binding with heparan sulfate. In the adhesion phase of HSV-1, the fundamental role is played by the glycoprotein gC, unlike HSV-2, which sees gB as the major contributor [19,20]. So, although the lipid composition of the two viruses is identical and is derived from the VERO cell membrane in both cases, the different arrangement of glycoproteins could account for the different sensitivity to KP, confirming a mechanism of action involving the attachment phase.

Overall, our results show a remarkable antiviral activity of KP against HSVs with different appealing features, such as the capability to also inhibit ACV-resistant strains, no selection of resistant mutants, and a synergistic action with ACV, suggesting that this AMP may represent a good therapeutic tool. Some issues, such as oral and systemic drug availability and the lack of effective delivery systems, still limit the clinical use of AMPs. However, the peculiar features of KP [11] suggest its potential, at least for a topical application in skin or eye lesions caused by HSVs.

## 4. Materials and Methods

### 4.1. Investigated Molecules

KP (AKVTMTCSAS) and a scrambled peptide (SP; MSTAVSKCAT) (m.w. 998.18) were synthesized by NeoMPS (PolyPeptide Group, Strasbourg, France). Peptide purity was 97.4% for KP and 95.8% for SP, as evaluated by analytical reverse phase HPLC. The peptides were solubilized in dimethylsulfoxide (DMSO) at a concentration of 20 mg/mL and subsequently diluted in phosphate-buffered saline (PBS) for experimental use. SP, characterized by the same amino acids as KP but in a different sequence, was used as a control. In all experiments, controls (without peptides) contained DMSO at the proper concentration.

ACV, purchased from Recordati (Recordati S.p.A., Milan, Italy), was resuspended with PBS. 

### 4.2. Viruses

HSV-1, HSV-2, and the ACV-resistant HSV isolates (ACVres-HSV-1 and ACVres-HSV-2) used in this study were clinical isolates identified by monoclonal antibodies. Wild-type HSV-1 and HSV-2 strains were laboratory-adapted through serial passages (>50) on Vero cells. Viral suspensions used as inocula were obtained from centrifuged lysates of virus-infected Vero cells cultured in a serum-free medium. Virus batches were titrated on Vero cells, aliquoted, and kept frozen at −80 °C; all the experiments were carried out with the same batch.

### 4.3. Cell Line

The Vero cell line was used in all experiments: it is part of the cell culture collection of the Laboratory of Virology of the University of Modena and Reggio, originally purchased from an Italian commercial cell bank. Vero cells were cultured at 37 °C in 5% CO_2_ atmosphere in Dulbecco′s modified Eagle′s medium (DMEM; PAN-Biotech, Aidenbach, Germany) with 10% (growth medium) or 5% (maintenance medium) Fetal Bovine Serum (FBS; GE Hyclone, Fisher Scientific Italia, Segrate, Italy), penicillin (100 U/mL; Euroclone, Pero, Italy), streptomycin (100 μg/mL; Euroclone), ciprofloxacin (100 μg/mL; Supelco), and L-glutamine (2 mM; Euroclone). Cells were maintained by bi-weekly passages in fresh medium. All the experiments were carried out in a medium without FBS in order to avoid interactions between serum components and peptides.

### 4.4. MTT Assay

The colorimetric 3-(4,5-dimethylthiazol-2-yl)-2,5-diphenyl tetrazolium bromide (MTT; Merck Life Science s.r.l., Milan, Italy) assay was used to evaluate the cytotoxicity of the investigated molecules on VERO cells and calculate the 50% cytotoxicity concentration (CC_50_), as previously described [21]. Briefly, cell cultures grown in monolayer (seeded one day earlier at 2 × 10^4^ cells/well in 96-well microplates) were exposed to different concentrations of KP, SP, and ACV ranging from 10 to 100 μg/mL, in triplicate, and incubated for 24 h at 37 °C. Then, the MTT assay was performed, and the cell viability was expressed as the optical density (OD) percentage of the treated cultures compared to that of the untreated controls (100% viability). CC_50_ was calculated by regression analysis as the concentration capable of reducing the OD in the MTT assay by 50% in comparison to control. 

### 4.5. Antiviral Assay

#### 4.5.1. Viral Yield Reduction Assay

The antiherpetic activity of KP was evaluated by the viral yield reduction assay. Vero cells grown in 96-well culture plates (seeded at 2 × 10^4^ cells/well concentration) for 24 h at 37 °C were infected with 25 μL of serum-free medium viral inoculum (10^4^ plaque forming units (PFU)/mL) with or without (as negative control) peptides at the concentrations of 10, 50, and 100 μg/mL. After 1 h adsorption at 37 °C, viral inocula were removed, and serum-free medium containing peptides at the same concentrations used for the infection or medium alone (control) was added. Each assay was performed in triplicate. After 24 h incubation at 37 °C, the supernatants were collected, and the virus yield was quantified by plaque reduction assay (PRA) [22]. From the infected cell cultures treated with 100 μg/mL of KP, the total DNA was extracted for quantification by real-time PCR (qPCR).

#### 4.5.2. Titration by Plaque Reduction Assay (PRA)

Vero cells were seeded in 24-well plates (1.5 × 10^5^ cells/well) and infected with 200 μL of 10-fold serial dilutions of each sample to be titrated. After 1 h incubation at 37 °C with 5% CO_2_, the inoculum was removed, and 200 μL of maintenance medium containing 0.6 % *v/v* human Ig (Ig Vena 50 g/L, Kedrion S.p.A., Castelvecchio Pascoli, Lucca, Italy) was added to each well. After 48–72 h of incubation, the cells were fixed with methanol and stained with Crystal Violet (CV). Plaques were scored visually, and the viral titer was expressed as PFU/mL. The viral inhibition was expressed as a Log of reduction in comparison with the negative control.

### 4.6. Viral Quantification by qPCR

Molecular quantification of viral growth reduction by KP (100 μg/mL) was carried out by real-time PCR. Viral DNA was extracted using a commercial kit (Wizard^®^ SV Genomic DNA Purification System; Promega S.r.l., Milan, Italy) according to the manufacturer’s instructions. *g*DNA concentration was determined by a NanoDrop™ 1000 spectrophotometer (Thermo Fisher Scientific, Segrate, Italy).

Primers for HSV-1 and HSV-2 DNA and β-globin gene (the Vero cells housekeeping gene) referred to literary data [23,24]. In both cases, β-genes were the primer targets: for the HSV-1, UL42 gene, and for HSV-2, the DNA-polymerase gene. The assay was performed in 96-well PCR plates using SYBR green (BioRad Laboratories S.r.l., Segrate, Italy). Fifty ng of *g*DNA was used for the reaction. DNA of infected and non-infected cells not treated with KP were used as controls. All samples were processed in triplicates.

The same amplification conditions were used for all genes: 5 min at 95 °C; 45 cycles of 5 s at 95 °C; and 30 s at primer specific temperature.

Ct values of viral target sequences were normalized to those of β-globin for Vero genes (Ct _HSV-1/2_ − Ct _β-globin_ = ∆Ct _sample_) in order to determine viral genome levels. ∆Ct values of mock-treated samples served as control, and relative quantification (RQ) was performed using the ∆∆Ct method (∆Ct _treated sample_ − ∆Ct _mock-treated sample_ = ∆∆Ct, RQ = 2^−∆∆Ct^). 

### 4.7. Time-of-Drug Addition Assay

This assay was performed on Vero cell monolayers in 96-well culture plates (seeded at 2 × 10^4^ cells/well concentration). The day after cell seeding, the cell cultures were infected with HSV for 1 h at 37 °C, as previously described, and then, the inoculum was removed, and cells were covered with serum-free medium and incubated for 24 h. KP at the concentration of 100 μg/mL was added to the wells in triplicate at different time points after infection (1, 3, 6, 12, and 18 h). Only for T0, KP was also added to viral inoculum, which was removed after 1 h, and KP-containing medium was added except in 3 wells (negative controls). At the same time, the viral inoculum was also removed from all the other wells and substituted with a serum-free medium, adding KP at different time points. Cell cultures were then incubated up to 24 h post-infection. Viral titer was determined in the supernatants by a PRA.

### 4.8. Evaluation of KP Virucidal Activity

Viral suspensions were added with KP or SP at the concentration of 100 μg/mL and incubated for 1 h at 37 °C before infecting the cell monolayers. The residual infectivity of the samples was then titrated by a PRA. 

### 4.9. Vero Cells Pretreatment

Vero cells monolayers seeded in 96-well microplates (2 × 10^4^ cells/well) were incubated for 1 h at 37 °C with or without (negative controls) KP or SP at the concentration of 100 μg/mL. Afterward, cells were washed with serum-free medium and infected with 25 μL of HSV, as previously described. After 1 h adsorption at 37 °C, viral inocula were removed, and serum-free medium without peptides was added. Each assay was performed in triplicate. After 24 h of incubation at 37 °C, viral titers were determined in the supernatants by a PRA. 

### 4.10. Inhibition of Adsorption Assay

Vero cells monolayers (seeded at 2 × 10^4^ cells/well concentration in 96-well microplates) were pre-incubated at 4 °C for 1 h and then infected with viral inocula containing 100 μg/mL of peptides (KP or SP). The samples were again incubated at 4 °C for 1 h and then washed twice with PBS and added with serum-free medium. Each assay was performed in triplicate. After 24 h of incubation at 37 °C, viral titers in the supernatants were determined by a PRA.

### 4.11. Inhibition of Penetration Assay

This test was performed as described by Gescher et al. [25], with minor modifications. Vero cells monolayers (seeded at 2 × 10^4^ cells/well in 96-well microplates) pre-incubated at 4 °C for 1 h were infected and further maintained at 4 °C for 1 h of adsorption. Afterward, wells were washed once with cold PBS (4 °C) to remove unattached viral particles, and serum-free medium was added with or without (negative controls) 100 μg/mL of peptides (KP or SP). The plates were incubated at 4 °C for 30 min and subsequently shifted to 37 °C to initiate viral penetration. After 30 min incubation, supernatants were discarded, and cells were treated for 45 s with low-pH citrate buffer (135 mM NaCL, 10 mM KCl, 40 mM Na-citrate pH 3.0) to stop virus penetration and inactivate attached but non-penetrated virions. Low-pH citrate buffer was removed by washing twice with PBS, and the maintenance medium was added. Each assay was performed in triplicate. After 24 h of incubation at 37 °C, viral titers in the supernatants were determined by a PRA.

### 4.12. Evaluation of Inhibitory Concentration 50 (IC_50_) and Selectivity Index (SI) of KP and ACV

The 50% inhibitory concentration (IC_50_) was established by PRA. Vero cells monolayers were infected with viral inocula containing two-fold compound dilutions (KP or ACV) at 37 °C for 1 h. Subsequently, the inocula were removed, and serum-free medium containing 0.6% *v/v* human Ig and KP or ACV at the same concentrations used in the inocula was added. After 24 h, a PRA was carried out. Each assay was performed in triplicate. IC_50_ was calculated through interpolation in a non-linear regression curve, as the concentration was capable of reducing the number of PFU by 50% in comparison with controls (untreated cells). The selectivity index (SI) was calculated as the ratio between the CC_50_ and the IC_50_ values.

### 4.13. Search for KP-Resistant Mutants

Several subcultures of both HSVs in the presence of a sub-inhibitory concentration of KP (10 μg/mL) were performed in order to isolate a KP-resistant HSV. This concentration was kept constant throughout the experiment. When a modest cytopathic effect was observed, viruses were subcultured to amplify the viral resistant population. For these passages, 100 μL of each amplification was used to generate the subsequent passage. After 10 and 20 subcultures, the sensitivity to KP was assessed by viral yield assay with or without KP (100 μg/mL), as described above. Viral titers were compared with those of the initial strains [26,27].

### 4.14. Synergy Evaluation

The synergistic effects of KP and ACV on HSV-1 were established by PRA in triplicate. Vero cells monolayers (seeded at 2 × 10^4^ cells/well concentration in 96-well microplates) were infected with viral inocula containing different 2-fold serial dilutions of KP or ACV. Each drug was added alone or in combination at a fixed ratio (1:1). Drug concentrations were chosen above and below their IC_50_ values. After incubation, the inocula were removed, and a serum-free medium containing 0.6% *v/v* human Ig and the same concentration of KP and ACV used in the inocula was added to each well. After 24 h of incubation, a PRA was carried out. The data obtained were expressed as a fractional inhibition (Fi) percentage. The synergistic effect of KP and ACV was calculated using a combination index (CI) through CompuSyn software, as described by Chou et al. The resulting CI theorem offers a quantitative definition for additive effect (CI = 1), synergism (CI < 1), and antagonism (CI > 1) in drug combinations [28,29]. A dose-reduction index (DRI) was calculated according to a previously described method [30].

### 4.15. Statistical Analysis

The data reported in the Figures are the mean values (±standard errors) from at least 3 different experiments analyzed using GraphPad PRISM^®^ 8 (GraphPad Software; San Diego, CA, USA). The results were analyzed by one-way ANOVA test with a Bonferroni post-hoc test. The level of significance was set at *p*-values < 0.05.

## Figures and Tables

**Figure 1 ijms-25-10602-f001:**
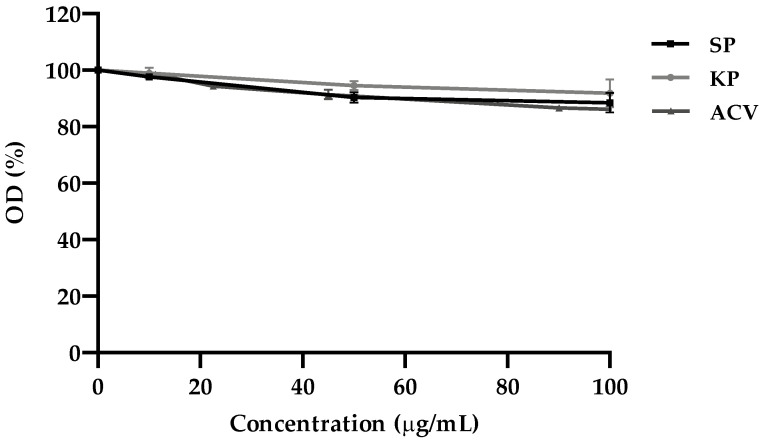
Cytotoxicity effects of KP, SP, and ACV on Vero cells. Influence of molecules on viability of uninfected Vero cells after 24 h of incubation was determined by MTT assay. The mean value of the optical density (OD) was calculated as a percentage of the treated cultures in comparison with the untreated controls (100% viability). Means ± SEM of three determinations are presented.

**Figure 2 ijms-25-10602-f002:**
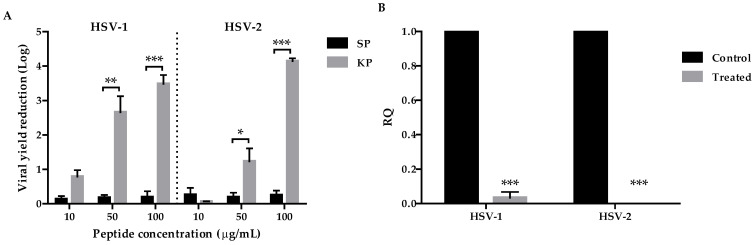
Antiviral activity of the investigated peptides. Vero cells were infected with HSV-1 or HSV-2 and treated with KP or SP (at 10, 50, and 100 μg/mL) added to cell culture medium during infection and maintained up to 24 h p.i. (**A**) Viral yield as determined by plaque reduction assay (PRA). The results are expressed as Log of reduction compared with the negative (untreated) controls. (**B**) HSV genome quantification was performed by real-time PCR from infected but untreated Vero cells (black columns, controls) and infected Vero cells treated with KP (100 μg/mL, gray columns). The results are expressed as relative viral load (RQ-values). The data represent the mean of three independent experiments ± SEM. Significant changes are marked with asterisks (* *p* < 0.05; ** *p* < 0.01; *** *p* < 0.001).

**Figure 3 ijms-25-10602-f003:**
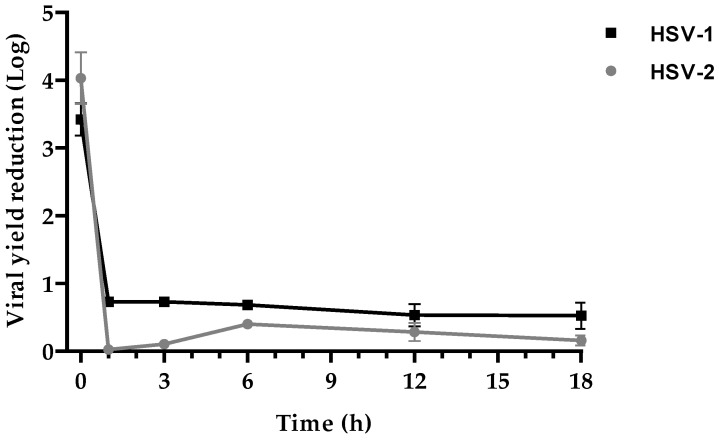
Time-of-drug addition assay. KP (100 μg/mL) was added to the cell culture concurrently with the infection (time 0) or at different time points (1, 3, 6, 12, 18 h) p.i. and was maintained in the culture medium up to 24 h. Viral yield was determined by PRA. Data are expressed as Log of reduction compared with the negative (untreated) controls and represent the mean of three independent experiments ± SEM.

**Figure 4 ijms-25-10602-f004:**
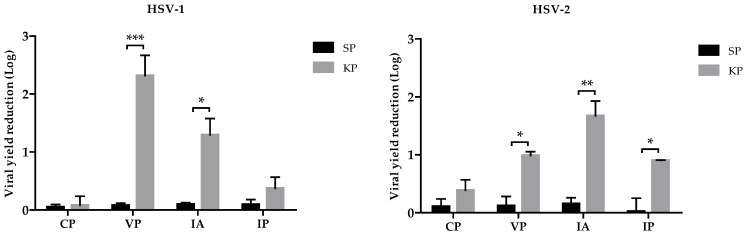
Antiviral effect of KP in early stages of HSV-1 and HSV-2 replicative cycle. Vero cells were infected with HSV under 4 different treatment conditions: (CP) Vero cells pretreatment; (VP) Virus pretreatment; (IA) Inhibition of adsorption assay; (IP) Inhibition of penetration assay. For each assay, viral titers were determined by PRA. See the Section 4 for methodological details. The data are expressed as Log of viral reduction compared with the negative (untreated) controls and represent the mean of three independent experiments ± SEM. Significant changes are marked with asterisks (* *p* < 0.05; ** *p* < 0.001; *** *p* < 0.0001).

**Figure 5 ijms-25-10602-f005:**
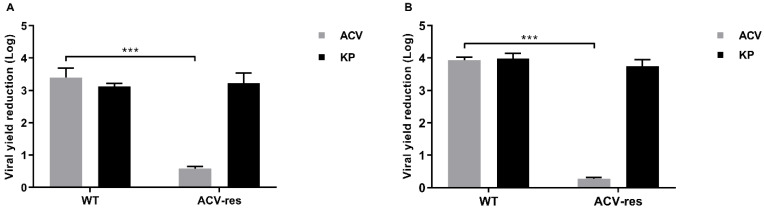
Antiviral activity of KP against ACV-resistant HSV isolates. See the Section 4 for methodological details. HSV-1 (A) and HSV-2 (B) wild type (WT) strains, ACV-resistant (ACV-res) isolates. The data represent the mean of three independent experiments ± SEM. Significant changes are marked with asterisks (*** *p* <0.0001).

**Figure 6 ijms-25-10602-f006:**
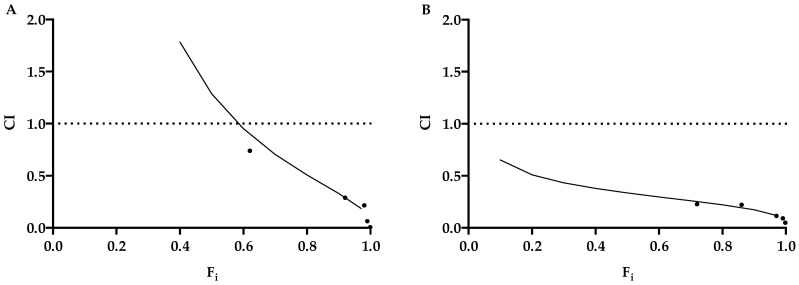
Fi-CI plots of KP and ACV interaction. Vero cells were infected with HSV-1 (**A**) and HSV-2 (**B**) and simultaneously treated with fixed combination ratios of KP and ACV. After 1 h of infection, cells were washed and overlaid with medium containing Ig and the mixture of compounds and incubated for 24 h. Then, the viral yield was quantified by PRA and expressed as Fractional inhibition (Fi), which represents the percent inhibition/100 determined on the mean of three replicates. In the Fi-CI plots, dots represent the CI value for a specific combination of KP and ACV concentrations. The dashed line (CI  =  1) indicates an additive reaction between the two substances. Values above and below this dashed line imply antagonism and synergism, respectively.

**Table 1 ijms-25-10602-t001:** Inhibitory Concentration 50 (IC_50_) and Selectivity Index (SI) values of KP and ACV against HSV-1 and HSV-2.

	CC_50_	HSV-1		HSV-2	
		IC_50_	SI	IC_50_	SI
KP	401.6	13.6	29.6	2.6	156
ACV	862.5	0.5	1742.5	1.3	669.1

CC_50_ and IC_50_ (both expressed as μg/mL) of KP and ACV for uninfected or HSV-infected Vero cells were determined by MTT and PRA, respectively. SI is the ratio between CC_50_ and IC_50_.

**Table 2 ijms-25-10602-t002:** Fractional inhibition, Combination Index, and Dose Reduction Index values of different concentrations of KP and ACV (ratio 1:1) against HSV-1 and HSV-2.

Virus	Fractional Inhibition (Fi)	Combination Index (CI)	Description	Dose Reduction Index (DRI)	
				ACV	KP
HSV-1	0.62	0.74	Moderate synergism	2.493	2.952
	0.92	0.29	Strong synergism	4.091	21.804
	0.98	0.214	Strong synergism	4.941	80.446
	0.99	0.065	Very strong synergism	15.471	2570.10
HSV-2	0.72	0.228	Strong synergism	1.572	3.612
	0.86	0.224	Strong synergism	3.131	7.786
	0.97	0.116	Strong synergism	11.650	33.689
	0.99	0.093	Very strong synergism	28.234	90.389
	0.998	0.050	Very strong synergism	101.505	376.423

DRI values >1 are beneficial, and the greater the DRI values, the greater the dose reduction for a given therapeutic effect.

## Data Availability

The raw data supporting the conclusions of this article will be made available by the authors on request.

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
