# Peer review of "Anti-Herpetic Activity of Killer Peptide (KP): An In Vitro Study"

_ijms, 2024, doi:10.3390/ijms251910602_

Round 1

Reviewer 1 Report

Comments and Suggestions for Authors

Please, see the file attached.

Author Response

Comments 1:

Line 52 ... immunocompromised patients has led to the selection of resistant viral strains...

Perhaps it is more correct to write ... development of resistant strains.

Response 1:

Thank you for this remark. We rephrased the sentence at line 53.

Reviewer 2 Report

Comments and Suggestions for Authors

In the manuscript submitted to me for review entitled "Anti-herpetic activity of Killer Peptide (KP): an in vitro study“ the authors Arianna Sala, Francesco Ricchi, Laura Giovati, Stefania Conti, Tecla Ciociola and Claudio Cermelli present a study of the anti-HSV-1 and HSV-2 activity of a broad-spectrum synthetic decapeptide, Killer Peptide (KP). The study examines the impact of different stages of viral replication and presents a future alternative for the treatment of viral infections through the use of antimicrobial peptides to avoid the side effects of therapy and the formation of resistant mutants.

The methods presented by the authors are well chosen and described and can be replicated in a subsequent study. The obtained results are presented in detail using 6 figures and 2 tables. To support their research, the authors used 27 references that present information from studies published mostly in the past three decades. About 1/2 of the total references are from the last 5 years, indicating that the development of new antiviral therapeutics has been a topic that has attracted investigator attention for years, but the increased occurrence of multiple virus epidemics in recent years has encouraged researchers to search for ever new treatment alternatives. This means that the present manuscript definitely presents information that would be of interest to IJMS readers. I did not notice any redundant self-citations, all the references used are appropriate and necessary for the preparation of the manuscript.

My remarks and recommendations to the authors are:

1. The description of figure 4 is very long. The description of the methodology itself can be removed. It is described in the Materials and Methods section.

2. The method part in the description of Figure 5 can also be omitted.

3. In section 2.7. an important characteristic of KP is clarified, namely that its use does not form resistant mutants. Data not presented. I think they are extremely important and it would be good if they were presented in some way or presented in an additional file.

4. As far as it is clear from the conducted experiments, the combined effect of KP and ACV was investigated against ACV-susceptible isolates. Why were the same experiments not performed on the resistant isolates? The main goal is to find additional and more effective therapy for patients with resistant strains. Why have similar experiments not been conducted with resistant isolates?

5. In section 4.2. reported that 4 types of virus isolates were used: 2 susceptible and 2 resistant to ACV therapy. In line 326 and elsewhere in the text, these isolates are called strains. The strains have a specific characteristic, are patented and named in a certain way. It is not very clear whether these are isolates or isolates transformed into strains by repeated passages although they are not named.

6. When describing the methodology and the results obtained from the Time-of-addition assay, the addition of KP in interval 0 is described. This is also the only interval in which a significant effect is reported. The way the experiment was conducted describes a real influence of KP only during the stage of virus adsorption and penetration into the cell. Isn't this actually affecting the adsorption and permeation step rather than the replication steps inside the cell? That is, the addition at hour 0 is similar to the study described in Section 4.10. The difference is that in the Time-of-addition assay the experiment was conducted at 37 °C and the penetration of virus particles took place, while in the Inhibition of adsorption assay at 4 °C it was limited only to the virus adsorption process.

7. How do methods 4.5.1 and 4.13 differ in the way they are conducted? Isn't the methodology itself the same regardless of whether the isolate is sensitive or resistant to ACV? In my opinion section 4.13. should be dropped.

Author Response

Comments 1:

The description of figure 4 is very long. The description of the methodology itself can be removed. It is described in the Materials and Methods section.

Response 1:

Thank you for your remarks. We have, accordingly, removed the methodology description from the figure 4 caption.

Comments 2:

The method part in the description of Figure 5 can also be omitted.

Response 2:

Thanks for the comment. We have omitted the description of methodology from the figure 5 caption.

Comments 3:

In section 2.7. an important characteristic of KP is clarified, namely that its use does not form resistant mutants. Data not presented. I think they are extremely important and it would be good if they were presented in some way or presented in an additional file.

Response 3:

Thank you for pointing this out. According to the reviewer’s suggestion, we introduced the description of these results in the text, considering the high number of figures/tables reported in the paper (lines 203 - 207 of the revised pdf manuscript with tracked changes).

Comments 4:

As far as it is clear from the conducted experiments, the combined effect of KP and ACV was investigated against ACV-susceptible isolates. Why were the same experiments not performed on the resistant isolates? The main goal is to find additional and more effective therapy for patients with resistant strains. Why have similar experiments not been conducted with resistant isolates?

Response 4:

The synergistic effect was deliberately studied only on ACV sensitive isolates, in the hypothesis that on the resistant isolates, we would have only seen the KP inhibitory activity.

Comments 5:

In section 4.2. reported that 4 types of virus isolates were used: 2 susceptible and 2 resistant to ACV therapy. In line 326 and elsewhere in the text, these isolates are called strains. The strains have a specific characteristic, are patented and named in a certain way. It is not very clear whether these are isolates or isolates transformed into strains by repeated passages although they are not named.

Response 5:

Thanks for the comment. The sensitive viruses used were originally clinically isolates which were in vitro cultured in our laboratory more than 20 years, so they are actually considered lab strains; oppositely, resistant viruses were isolated from patients with therapeutic failure and had been subjected to a very limited number of passages on Vero cells before the experiments. We accordingly changed the terms in the paper, maintaining strain for the sensitive ones and isolates for the resistant ones.

Comments 6:

When describing the methodology and the results obtained from the Time-of-addition assay, the addition of KP in interval 0 is described. This is also the only interval in which a significant effect is reported. The way the experiment was conducted describes a real influence of KP only during the stage of virus adsorption and penetration into the cell. Isn't this actually affecting the adsorption and permeation step rather than the replication steps inside the cell? That is, the addition at hour 0 is similar to the study described in Section 4.10. The difference is that in the Time-of-addition assay the experiment was conducted at 37 °C and the penetration of virus particles took place, while in the Inhibition of adsorption assay at 4 °C it was limited only to the virus adsorption process.

Response 6:

Yes, the Referee got the point. The different temperatures for assay execution depends on the fact that while virus adsorption is a passive step not requiring cell metabolic activity, so it can occur also at 4° C, virus entry needs an active participation of cell machinery, at 37° C. Working with two different temperatures allows researchers to discriminate the event involved. We added this clarification in the text (lines 280 - 284 of the revised pdf manuscript with tracked changes).

Comments 7:

How do methods 4.5.1 and 4.13 differ in the way they are conducted? Isn't the methodology itself the same regardless of whether the isolate is sensitive or resistant to ACV? In my opinion section 4.13. should be dropped.

Response 7:

We agree with this comment, the section 4.13 is redundant, and we deleted.

Reviewer 3 Report

Comments and Suggestions for Authors

The manuscript by Sala et al. addresses an important topic by investigating the antiviral effects of Killer Peptide (KP) against HSV-1 and HSV-2, offering potential alternative treatments for drug-resistant strains, which is highly relevant. The finding of synergistic effects between KP and acyclovir (ACV) is particularly notable, as it suggests the possibility of combination therapies that could reduce both toxicity and cost, as the authors have claimed.

The manuscript is generally well-written and well-organized, with the conclusion supported by the data. However, there a few significant concerns that need to be addressed:

Comments and Concerns

 #1) In the viral genome quantification (Fig. 2B) and time-of-drug addition assay (Fig. 3), the authors did not include ACV as a positive control. Since the study compares KP with ACV throughout, ACV should also be included in these assays to develop a better understanding of the relative effectiveness of KP.

#2) The authors did not specify whether the quantified viral genes in the real-time PCR were early, intermediate, or late genes. This omission could affect the interpretation of the viral load during different stages of infection. The authors should clarify which viral genes were targeted in the qPCR assays.

#3) The authors mentioned that clinical HSV-1 and HSV-2 strains were used, but they did not specify their lineage (e.g., Mcrae, KOS,). Mentioning and referencing the specific strains will improve the clarity of the study, especially when interpreting its broader applicability.

#4) Some methods, such as those used for adapting ACV-resistant strains, are not well-documented. In Section 4.2 (Materials and Methods), the authors should provide more detail on how the resistant strains were adapted. Additional references for these methods would also be beneficial.

#5) The discussion section should elaborate on how KP impacts specific glycoproteins, such as gC and gB, in HSV-1 and HSV-2. Providing more detail in this regard will strengthen the mechanistic understanding the antiviral action of KP.

Minor Points

Lines, 114 and 122: the word (addiction) should be corrected to (addition).

Lines 294-302: The authors should cite references to support this part of the discussion.

Author Response

Comments 1:

In the viral genome quantification (Fig. 2B) and time-of-drug addition assay (Fig. 3), the authors did not include ACV as a positive control. Since the study compares KP with ACV throughout, ACV should also be included in these assays to develop a better understanding of the relative effectiveness of KP.

Response 1:

The aim of time-of-drug addition assay is to understand in which phase of the virus replication cycle a drug exerts its action. The mechanism of action of ACV is well known as well its molecular targets, therefore we did not perform this test for ACV.

Comments 2:

The authors did not specify whether the quantified viral genes in the real-time PCR were early, intermediate, or late genes. This omission could affect the interpretation of the viral load during different stages of infection. The authors should clarify which viral genes were targeted in the qPCR assays.

Response 2:

Thanks for this remark which gives us the chance to clarify this point in the text. We did not specify the targets of the primers used considering that the aim of the quantitative PCR was to confirm the results of the biological test, the plaque reduction assay, less precise and sensitive, and not to understand the mechanism(s) of action of KP for which we conducted other tests. Anyway, we specified in the test the primer targets (beta-genes for both viruses) (lines 392-393 of the revised pdf manuscript with tracked changes).

Comments 3:

The authors mentioned that clinical HSV-1 and HSV-2 strains were used, but they did not specify their lineage (e.g., Mcrae, KOS,). Mentioning and referencing the specific strains will improve the clarity of the study, especially when interpreting its broader applicability.

Response 3:

Thank you for pointing this out. We agree that the lineage is an important feature, but unfortunately, we do not have this information. The wild type strains used were originally clinically isolates which were in vitro cultured in our laboratory more than 20 years, originally identified by IFA with specific mAbs; similarly, resistant viruses were recently isolated from patients with therapeutic failure, identified by clinicians with a PCR assay but subjected to a very limited number of passages on Vero cells before the experiments. However, as far as we know from the literature, different lineages usually display different pathogenic features rather than drug sensitivity.

Comments 4:

Some methods, such as those used for adapting ACV-resistant strains, are not well-documented. In Section 4.2 (Materials and Methods), the authors should provide more detail on how the resistant strains were adapted. Additional references for these methods would also be beneficial.

Response 4:

Thank you for your remarks. More detail and additional references were added in Sections 4.2 and 4.13 of the revised manuscript. The added the references are numbers 26 and 27 of the revised manuscript.

Comments 5:

The discussion section should elaborate on how KP impacts specific glycoproteins, such as gC and gB, in HSV-1 and HSV-2. Providing more detail in this regard will strengthen the mechanistic understanding the antiviral action of KP.

Response 5:

The referee’s comment is right. However, the aim of this first work was to ascertain the anti-herpetic activity of KP and suggest possible mechanisms of action. We are currently carrying out an in silico study of the molecular interactions of KP with all the envelope glycoproteins.

Comments 6:

Lines, 114 and 122: the word (addiction) should be corrected to (addition).

Response 6:

Sorry for these mistakes. We changed these words.

Comments 7:

Lines 294-302: The authors should cite references to support this part of the discussion.

Response 7:

According to the reviewer’s suggestion, we have cited references to support the Discussion. The added the references are numbers 19 and 20 of the revised manuscript.

Round 2

Reviewer 3 Report

Comments and Suggestions for Authors

All comments and concerns have been appropriately addressed, and I recommend the manuscript for publication.